# Multi-Class Positive and Unlabeled Learning for High Dimensional Data Based on Outlier Detection in a Low Dimensional Embedding Space

Cheong Hee Park 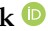

Division of Computer Convergence, Chungnam National University, Daejeon 34134, Korea; cheonghee@cnu.ac.kr;
Tel.: +82-42-821-6293

**Abstract:** Positive and unlabeled (PU) learning targets a binary classifier on labeled positive data and unlabeled data containing data samples of positive and unknown negative classes, whereas multi-class positive and unlabeled (MPU) learning aims to learn a multi-class classifier assuming labeled data from multiple positive classes. In this paper, we propose a two-step approach for MPU learning on high dimensional data. In the first step, negative samples are selected from unlabeled data using an ensemble of *k*-nearest neighbors-based outlier detection models in a low dimensional space which is embedded by a linear discriminant function. We present an approach for binary prediction which determines whether a data sample is a negative data sample. In the second step, the linear discriminant function is optimized on the labeled positive data and negative samples selected in the first step. It alternates between updating the parameters of the linear discriminant function and selecting reliable negative samples by detecting outliers in a low-dimensional space. Experimental results using high dimensional text data demonstrate the high performance of the proposed MPU learning method.

**Keywords:** high dimensional data; *K* nearest neighbors-based outlier detection; low dimensional embedding; multi-class positive and unlabeled learning; positive and unlabeled learning

## 1. Introduction

A classification problem requires training data along with class labels to learn a classifier, and in general, the test data are expected to come from the same distribution as the training data. However, it can be difficult to have labeled data for all classes. Positive and unlabeled (PU) learning aims to learn a classifier when labeled data from a positive class and unlabeled data from both positive and unknown negative classes are given [1,2]. While PU learning is based on a binary classification, multi-class positive and unlabeled (MPU) learning assumes that labeled data from multiple positive classes and unlabeled data from either the positive classes or an unknown negative class are given [3,4]. MPU learning can arise in various real problems. In fraud transaction detection where fraud transaction data can be considered as positive data, there are multiple fraud types [4], and in document classification, labeled positive data can be composed of several categories of documents.

While many PU learning methods exist [1,2,5], the MPU learning method was first proposed in [3]. In [3], different loss functions for labeled data and unlabeled data are constructed to eliminate estimation bias. The original data space is mapped to an embedding space where the codewords corresponding to each class are fixed by a maximum margin. The parameter matrix of a linear discriminant function and the label estimation of unlabeled data samples are alternately optimized. In [4], to address the overfitting problem caused due to the unbounded risk of the method of [3], an alternative risk estimator with the modification of the hinge loss function has been proposed. The methods in [3,4] require the estimation of class priors which are mainly used in the construction of loss functions, but the exact estimation of class priors is difficult when no negative labels are given.



Many PU learning methods adopt a two-step approach to identify reliable negative samples and learn a binary classifier [6–9]. In [10], unlabeled samples are ranked using outlier scores calculated as the sum of the distances to the $k$-nearest positive samples, and unlabeled samples with outlier scores above a threshold are selected as negative samples. The $k$-nearest neighbors ($k$nn)-based outlier detection method is simple and known to be comparable to other popular outlier detection methods. However, in high dimensional data, the curse of dimensionality can cause data sparsity and equidistance between data samples. Moreover, setting a threshold on $k$nn-based outlier scores for negative sample selection is not trivial.

In this paper, we propose a method for multi-class positive and unlabeled (MPU) learning on high dimensional data based on outlier detection in low dimensional embedding space. The original data space is mapped to a low dimensional space by a linear discriminant function, and reliable negative data samples are selected by using an ensemble of $k$nn-based outlier detection models in low dimensional data space. We present an approach for binary prediction of outliers that determines whether a data sample is a negative sample or not. Parameter updates of a linear discriminant function and the selection of reliable negative samples by outlier detection in low dimensional space are alternately performed. Experimental results using high dimensional text data demonstrate the high performance of the proposed MPU learning method.

The rest of the paper is organized as follows. In Section 2, related work is reviewed, and in Section 3, we present a method for MPU learning based on outlier detection. Experimental results which apply MPU learning for text classification are given in Section 4 and the conclusion follows in Section 5.

## 2. Related Work

PU learning targets a binary classifier on labeled data from a positive class and unlabeled data either from positive or unknown negative classes. PU learning methods can be divided into three categories [2,4]: (1) Use a two-step approach of identifying reliable negative examples in unlabeled data, and learning a classifier based on positives and identified reliable negatives. (2) Consider unlabeled data as negative data with label noise. (3) Consider unlabeled data negative and formulate the problem as a cost-sensitive learning problem. Extensive surveys for PU learning methods and application problems can be found in [1,2,5].

While most PU learning methods are limited to binary classification, a method for multi-class positive and unlabeled learning was proposed in [3]. Suppose that labeled data from $m - 1$ positive classes and unlabeled data from either positive classes or an unknown negative class are given. When the negative class is denoted as the $m$-th class, let $z_i \in R^{m-1}$, $i = 1, \ldots, m$, be codewords for each class where the margins between them are maximized. A linear discriminant classifier $f$ was defined as $f(x) = argmax_{1 \leq i \leq m} (Wx)^T z_i$ for the parameter matrix $W \in R^{(m-1) \times d}$ and $x \in R^d$. Given the class prior $\pi_i$, $i = 1, \ldots, m$, the method in [3] defined the objective function to use distinct loss functions for labeled and unlabeled data such as

$$
\begin{aligned}
J(W, \widetilde{y}) = {} & \frac{\lambda}{2} \|W\|_F^2 + \frac{1}{m-1} \sum_{i=1}^{m-1} \frac{\pi_i}{2|C_i|} \sum_{x \in C_i} \max\{0, (Wx)^T (z_k - z_i)\} \\
& + \frac{1}{m-1} \sum_{i=1}^{m-1} \frac{1}{2|C_u|} \sum_{x \in C_u} \max\{0, 1 + (Wx)^T (z_i - z_{\widetilde{y}_x})\}.
\end{aligned}
\tag{1}
$$

$C_i$ and $C_u$ denote the set of data samples in class $i$, $(1 \leq i \leq m - 1)$, and the set of unlabeled data samples, respectively, and $\widetilde{y}_x$ is the predicted class label for unlabeled data x. The first term in Equation (1) is the regularization term, and the second and third terms are the loss of labeled and unlabeled data. The optimization of $W$ and the estimation of $\widetilde{y}$ are alternated by fixing one of them. However, as pointed out in [4], the incorrectly estimated pseudo-labels $\widetilde{y}$ of the unlabeled data may disrupt the subsequent model training.

In [4], to prevent the unbounded risk caused by $-h(x)$ where h is the binary loss such as hinge loss, the substitution of $-h(x)$ by $h(-x)$ was proposed, and the empirical loss was defined as

$$
\begin{aligned}
J(f) = \frac{m}{m-1} \sum_{i=1}^{m-1} \frac{\pi_i}{|C_i|} \sum_{x \in C_i} [h(f_i(x)) + h(-f_m(x))] \\
+ \frac{1}{|C_u|} \sum_{x \in C_u} \left[ h(f_m(x)) + \frac{1}{m-1} \sum_{i=1}^{m-1} h(-f_i(x)) \right]
\end{aligned}
\tag{2}
$$

for hinge loss $h$ and $m$ linear classifiers $f_i(\mathbf{x}) = w_i^T \mathbf{x}$. The gradient descent method was used for the optimization of Equation (2).

### 3. The Proposed MPU Learning Method Based on Outlier Detection in a Low Dimensional Embedding Space

We propose a two-step approach for multi-class positive and unlabeled learning on high dimensional data. In the first step, negative samples are selected from unlabeled data using an ensemble of knn-based outlier detection models in a low dimensional space embedded by a linear discriminant function. In the second step, the linear discriminant function is optimized using hinge loss or squared errors on multiple positive classes and the negative class of samples selected in the first step. This two-step process is repeated. In the following subsections, we explain the proposed method in detail.

#### 3.1. Selection of Negative Samples Based on Binary Outlier Detection

It is known that outlier detection based on $k$-nearest neighbor distance is comparable to well-known outlier detection methods such as Isolation Forest or autoencoder-based methods [11]. It computes outlier scores using the maximum or average of the distances from a test sample to $k$-nearest neighbors. The farther the data sample is from the normal data region, the greater the outlier score. In [10], an outlier score was computed for each unlabeled data sample by the sum of the distances to its $k$-nearest neighbors among positive data samples, and a threshold on outlier scores was set for the selection of negative samples in unlabeled data. However, it is not reliable to use an arbitrary threshold, since the selection of negative samples makes a significant impact on the performance of a classifier in PU or MPU learning.

As the first step for multi-class positive and unlabeled learning, we compose an ensemble of $k$nn-based outlier detection models and set different thresholds for each outlier detection model. Suppose the labeled data are from positive classes, 1, ..., $m - 1$. An ensemble of $k$nn-based outlier detection models is constructed by building an outlier detection model for each positive class $i$, $(1 \leq i \leq m - 1)$, where $k$-nearest neighbors are searched in that positive class. We determine a threshold $s_i$ for the outlier detection model of class $i$ by composing a training set and validation set as follows:

- We construct the training set $T_i$ by randomly selecting 90% of the data samples from class $i$ and set $V_{ii}$ as the remaining 10% of data samples from class $i$.
- For each positive class $j$ except $i$, 10% of the data samples are randomly selected to construct a set $V_{ij}$, and then a validation set $V_i$ is constructed with $V_i = \mathrm{U}_{j=1}^{m-1} V_{ij}$.
- For each data sample in $V_i$, compute the sum of distances to $k$-nearest neighbors in the training set $T_i$ as the outlier score.
- Repeat the following procedure for $p$ = 80, 85, 90, 95, 100 and determine the $p$-th percentile with the highest f1 score as the threshold $s_i$ for prediction to class $i$.
  - When setting a threshold as the $p$-th percentile of the outlier scores of the data samples in $V_{ii}$, if the outlier scores of the data samples in $V_i$ are less than the threshold, then we predict that they belong to class $i$. The f1 score is computed for the prediction of the data samples in $V_i$ to class $i$.
- After constructing $T_i$ and $s_i$ for each positive class $i$, the process of selecting negative samples from unlabeled data is as follows: For unlabeled data sample $x$, if the sum of distances to $k$-nearest neighbors among data samples in $T_i$ is greater than $s_i$, then $x$ is

declared not to belong to class *i*. When *x* does not belong to any positive classes, it is chosen as the negative-class sample.

### 3.2. Learning a Multi-Class Classifier

A linear discriminant function represents a separating hyperplane between two classes [12]. For a multi-class problem, a classifier composed of linear discriminant functions $\{f_i | i = 1, \ldots, m\}$ assigns a data sample to class *i* such that $i = argmax_{1 \leq j \leq m} f_j(x)$. For multi-class positive and unlabeled learning, we learn a linear classifier $f = [f_1, \ldots, f_m]$ for $m - 1$ positive classes and one negative class which maps $x \in R^d$ to $f(x) \in R^m$ and define an objective function using hinge loss. The objective function by the hinge loss $h(z) = max\{0, 1 - z\}$ can be defined as

$$\text{minimize } \sum_{i=1}^{m} \frac{1}{|C_i|} \sum_{x \in C_i} [h(f_i(x)) + \sum_{j=1, j \neq i}^{m} h(-f_j(x))] \tag{3}$$

where $C_i$ is the set of data samples in class *i* and $n_i$ is the size of $C_i$. In the experiments of Section 4, we implemented the discriminant function *f* by a linear neural network with no hidden layer and trained the model using the gradient descent method.

Data representation in the output layer mapped by a linear neural network can provide improved class separability such as in [13]. Hence, we perform the selection of negative samples in the space transformed by *f* instead of the original data space. Moreover, the selection of more reliable negative samples is pursued by performing alternately the update of weight parameters of the linear neural network and the selection of negative samples at every epoch in training the network. Algorithm 1 summarizes the proposed multi-class positive and unlabeled learning method based on negative sample selection by outlier detection in a transformed low dimensional space. Algorithm 2 describes the multi-positive and unlabeled learning method based on negative sample selection by outlier detection in the original data space.

---

**Algorithm 1.** The proposed MPU learning algorithm based on outlier detection in the projected lower dimensional space

---

Input P: labeled positive data from class $i = 1, \ldots, m - 1$.
U: unlabeled data containing positive and negative data

1. Initialize the weights in a linear neural network with *d* nodes in an input layer and m nodes in the output layer implementing the discriminant function *f*.
2. Repeat
3. Compute $f(P \cup U)$.
4. Compose the set of negative samples, $N \subset U$, by performing the method presented in Section 3.1 on the data $f(P \cup U)$
5. Update weights of the network by the gradient descent method for the objective function in Equation (3) using training data $P \cup N$
6. Until the stop condition meets

---

**Algorithm 2.** The proposed MPU learning algorithm based on outlier detection in the original data space

---

Input P: labeled positive data from class $i = 1, \ldots, m - 1$.
U: unlabeled data containing positive and negative data

1. Compose the set of negative samples, $N \subset U$, by performing the method presented in Section 3.1 on $P \cup U$
2. Initialize the weights of a linear neural network with *d* nodes in an input layer and *m* nodes in the output layer.
3. Repeat
4. Update weights of the network by the gradient descent method for the objective function in Equation (3) using training data $P \cup N$
5. Until the stop condition meets

## 4. Experiments

### 4.1. Data Description

We used nine text data sets for performance comparison which have been used in the experiments on high dimensional data [14], and a detailed description is given in Table 1. The BBC News data was preprocessed to have 17,005 terms by deleting special symbols and numbers and removing terms appearing in only one document [15]. Reuters-21578 was downloaded from the UCI machine learning repository and the documents belonging to 135 TOPICS categories were used. After preprocessing by stopwords removal, stemming, tf–idf transformation, and unit norm, and excluding documents belonging to two or more categories, there were 6,656 documents composed of 15,484 terms, and the two largest categories of 1 and 36 and the collection of remaining documents composed three classes [14]. The 20-newsgroup (20-ng) data contained about 20,000 articles in 20 news groups divided into five categories [16]. After preprocessing the 20news-bydate version, we constructed 18,774 text data with 44,713 terms. Medline data is a subset of the MEDLINE database with five classes. Each class has 500 documents. After stemming and stoplist removal, it contained 22,095 distinct terms [17]. The remaining five data sets were downloaded from [18]. They were constructed by removing classes with less than 200 texts and terms with frequencies less than or equal to 1 [14].

**Table 1.** Data description.

| Data | Terms | Samples | Classes |
|---|---|---|---|
| bbc | 17,005 | 2225 | 5 |
| reuter | 15,484 | 6656 | 3 |
| 20-ng | 44,713 | 18,774 | 20 |
| medline | 22,095 | 2500 | 5 |
| la12 | 21,604 | 6279 | 6 |
| sports | 18,324 | 8313 | 5 |
| classic | 12,009 | 7094 | 4 |
| ohscal | 11,465 | 11,162 | 10 |
| reviews | 23,220 | 3932 | 4 |

### 4.2. Experimental Setup

The MPU learning problem was simulated as follows. In each data set, half of the classes were randomly chosen and served as multiple positive classes. From each of those classes, 40% of the data samples were randomly selected to compose labeled positive data, another 40% of the data samples were for unlabeled positive data, and the remaining 20% of the data samples composed test data. Data samples of the remaining half of the classes composed one negative class. From each of those classes, 40% of the data samples were randomly selected as unlabeled data of the negative class, and 20% of the data samples as test data of the negative class. A classifier for $m$ classes ($m - 1$ positive classes and one negative class) was trained by MPU learning methods using labeled and unlabeled data and test data accuracy was measured. For each data set, we repeated this procedure 20 times by randomly splitting it into labeled, unlabeled, and test data, and the average accuracy is reported in Table 2. For 20-newsgroup data, the classes in one category among five categories are served as positive classes; data of the classes in another category compose one negative class, and random category selection was repeated 20 times.

**Table 2.** Performance comparison by accuracy and F1-measure.

| Data | Accuracy in Multi-Class Positive vs. Negative | | | | | F1-Measure in Positive vs. Negative | | | |
|---|---|---|---|---|---|---|---|---|---|
| | Paper [3] | Paper [4] | U-Neg | Algo2 | Algo1 | nnPU [19] | U-Neg | Algo2 | Algo1 |
| bbc | 0.939 | 0.851 | 0.941 | 0.955 | 0.961 | 0.892 | 0.961 | 0.961 | 0.975 |
| reuter | 0.885 * | 0.742 | 0.872 | 0.928 | 0.939 | 0.943 | 0.933 | 0.959 | 0.969 |
| 20-ng | * | 0.63 | 0.886 | 0.899 | 0.903 | 0.773 | 0.941 | 0.949 | 0.953 |
| medline | 0.616 | 0.816 | 0.812 | 0.811 | 0.833 | 0.811 | 0.891 | 0.891 | 0.903 |
| la12 | 0.808 | 0.69 | 0.833 | 0.861 | 0.874 | 0.625 | 0.862 | 0.882 | 0.894 |
| sports | 0.887 * | 0.721 | 0.968 | 0.973 | 0.973 | 0.903 | 0.976 | 0.979 | 0.978 |
| classic | 0.929 | 0.942 | 0.944 | 0.941 | 0.959 | 0.771 | 0.95 | 0.944 | 0.964 |
| ohscal | * | 0.603 | 0.575 | 0.655 | 0.668 | 0.732 | 0.736 | 0.78 | 0.788 |
| reviews | 0.839 | 0.73 | 0.899 | 0.88 | 0.918 | 0.73 | 0.91 | 0.897 | 0.928 |
| average | 0.843 | 0.747 | 0.859 | 0.879 | 0.892 | 0.787 | 0.907 | 0.917 | 0.928 |

\* cases where the repetition of 20 times was not fulfilled due to the shortage in memory allocation or other errors.

We compared the performance of the following MPU learning methods.

- Paper [3]: Different convex loss functions are constructed for labeled and unlabeled data as in Equation (1), and the optimization of a discriminant function and the prediction of unlabeled data are performed alternately.
- Paper [4]: A risk estimator bounded below is proposed by using the modification of the hinge loss function as in Equation (2).
- U-Neg: This simple and naive approach assumes that the unlabeled data all belong to the negative class, commonly referred to as the closed-world assumption [2].
- Algo1: The proposed MPU learning method which performs outlier detection in a low dimensional embedding space.
- Algo2: The proposed MPU learning method which performs outlier detection in the original data space.

In addition, the performance of a positive and unlabeled learning method [19] for binary classification on labeled positive data and unlabeled negative data is also compared.

- nnPU [19]: A non-negative risk estimator for positive and unlabeled(PU) learning that explicitly constrains the training risk to be non-negative.

All the methods were implemented in Python. For the method in [3], we used the CVXPY package which is an open source Python-embedded modeling language for convex optimization problems [20]. However, the number of terms in text data is very high as shown in Table 1, and for some datasets, it failed to allocate memory for matrix parameters of Equation (1). The cases where the repetition of 20 times was not completed due to the shortage in memory allocation or other errors in a training stage were marked with an asterisk in Table 2. We used PyTorch [21] for the method of [4] and the proposed method. Wherever possible, the default values of the hyper-parameters provided in the reference papers were used: for the method [4] and the proposed method, learning rate = 0.01, weight decay = 0.01, initialization of weights by Xavier uniform method, optimizer = Adam optimizer, mini-batch size = 256, epochs = 11, $k = 5$ in knn. The number of epochs was set as 11 and training loss was used as a stop condition. For the method [19], the implementation code (https://github.com/kiryor/nnPUlearning, accessed on 16 July 2022) and the default parameter values by the authors were used, except for a base classifier. Instead of using the default base classifier, we tested all three base classifiers implemented in the code, a linear neural network, a two-layered neural network, and a five-layered neural network, and the best of them was used as the final result.

Figure 1 shows a simplified example of MPU learning and PU learning. In the example for MPU learning in the left column, the accuracy is the proportion of data samples

predicted correctly in the test data. For comparison with the PU learning method, nnPU [19], we also computed the F1-measure which is an evaluation measure for binary classification of positive and negative classes. As in the example in the right column, nnPU is applied after setting multi-class positive data as one positive class.

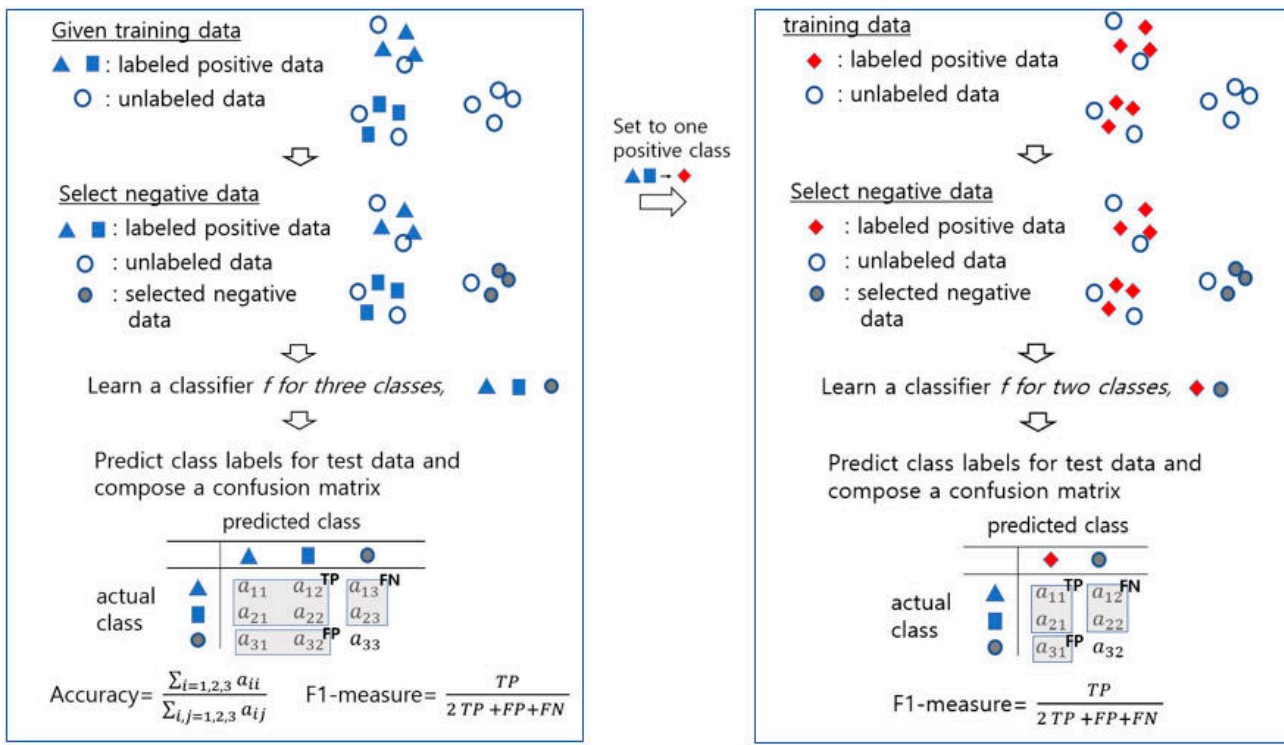

MPU learning                                                    PU learning

**Figure 1.** An example of MPU learning and PU learning.

### 4.3. Performance Comparison

Table 2 compares the accuracy of MPU learning methods and F1-measure for comparison with the PU learning method, nnPU. As shown in Table 2, the proposed method showed the highest performance among the compared methods. In particular, when outlier detection was performed in low dimensional embedding space, the average accuracy was improved compared with that when outlier detection was performed in the original data space. It is also noteworthy that the simple method of assuming unlabeled data as negative data performed comparably to the more complicated method.

Next, we repeated the same experiment varying the number of unlabeled negative data samples. In the previous experiment, from each class not selected as a positive class, 40% of data samples were randomly selected as unlabeled negative data. This time we constructed unlabeled negative data by randomly selecting 20% or 80% of the data samples from each class not selected as a positive class and performed the same experiment. Tables 3 and 4 show the average accuracy and F1-measure in two cases, and Figure 2 shows the aggregated average values of the MPU training methods in the last row of Tables 2–4. The performance of the proposed MPU learning method was stable regardless of the amount of unlabeled negative data. The difference in accuracy between Algorithm 1 and the method in [3] increased from 2% to 9% as the amount of unlabeled negative data increased, as shown in Tables 3 and 4.

**Table 3.** Performance comparison by accuracy and F1-measure when 20% of data samples randomly selected from each class constitute unlabeled negative data.

| Data | Accuracy in Multi-Class Positive vs. Negative | | | | | F1-Measure in Positive Vs. Negative | | | |
|---|---|---|---|---|---|---|---|---|---|
|  | Paper [3] | Paper [4] | U-Neg | Algo2 | Algo1 | nnPU [19] | U-Neg | Algo2 | Algo1 |
| bbc | 0.903 | 0.636 | 0.894 | 0.943 | 0.946 | 0.806 | 0.926 | 0.961 | 0.963 |
| reuter | 0.866 | 0.718 | 0.81 | 0.924 | 0.937 | 0.924 | 0.897 | 0.956 | 0.967 |
| 20-ng | 0.775 * | 0.67 | 0.859 | 0.892 | 0.902 | 0.855 | 0.92 | 0.943 | 0.953 |
| medline | 0.544 | 0.563 | 0.769 | 0.789 | 0.802 | 0.766 | 0.864 | 0.879 | 0.883 |
| la12 | 0.775 | 0.569 | 0.767 | 0.836 | 0.861 | 0.738 | 0.813 | 0.862 | 0.883 |
| sports | 0.913 | 0.581 | 0.941 | 0.968 | 0.968 | 0.906 | 0.957 | 0.974 | 0.975 |
| classic | 0.875 | 0.631 | 0.877 | 0.927 | 0.947 | 0.856 | 0.895 | 0.932 | 0.951 |
| ohscal | 0.639 * | 0.581 | 0.511 | 0.631 | 0.638 | 0.744 | 0.703 | 0.766 | 0.77 |
| reviews | 0.798 | 0.592 | 0.83 | 0.865 | 0.902 | 0.771 | 0.861 | 0.881 | 0.914 |
| average | 0.788 | 0.616 | 0.806 | 0.864 | 0.878 | 0.818 | 0.871 | 0.906 | 0.918 |

\* cases where the repetition of 20 times was not fulfilled due to the shortage in memory allocation or other errors.

**Table 4.** Performance comparison by accuracy and F1-measure when 80% of data samples randomly selected from each class constitute unlabeled negative data.

| Data | Accuracy in Multi-Class Positive vs. Negative | | | | | F1-Measure in Positive vs. Negative | | | |
|---|---|---|---|---|---|---|---|---|---|
|  | Paper [3] | Paper [4] | U-Neg | Algo2 | Algo1 | nnPU [19] | U-Neg | Algo2 | Algo1 |
| bbc | 0.952 | 0.684 | 0.965 | 0.965 | 0.97 | 0.686 | 0.979 | 0.978 | 0.982 |
| reuter | 0.893 | 0.761 | 0.922 | 0.928 | 0.94 | 0.94 | 0.962 | 0.959 | 0.97 |
| 20-ng | * | 0.53 | 0.898 | 0.901 | 0.903 | 0.723 | 0.95 | 0.951 | 0.954 |
| medline | 0.666 | 0.499 | 0.848 | 0.828 | 0.853 | 0.306 | 0.914 | 0.903 | 0.917 |
| la12 | * | 0.684 | 0.869 | 0.866 | 0.881 | 0.525 | 0.891 | 0.886 | 0.9 |
| sports | 0.932 * | 0.79 | 0.979 | 0.977 | 0.976 | 0.869 | 0.985 | 0.982 | 0.981 |
| classic | 0.954 | 0.877 | 0.963 | 0.952 | 0.963 | 0.852 | 0.968 | 0.954 | 0.968 |
| ohscal | * | 0.504 | 0.628 | 0.672 | 0.685 | 0.749 | 0.765 | 0.79 | 0.797 |
| reviews | 0.857 | 0.755 | 0.929 | 0.896 | 0.929 | 0.606 | 0.936 | 0.898 | 0.937 |
| average | 0.876 | 0.676 | 0.889 | 0.886 | 0.9 | 0.695 | 0.928 | 0.922 | 0.934 |

\* cases where the repetition of 20 times was not fulfilled due to the shortage in memory allocation or other errors.

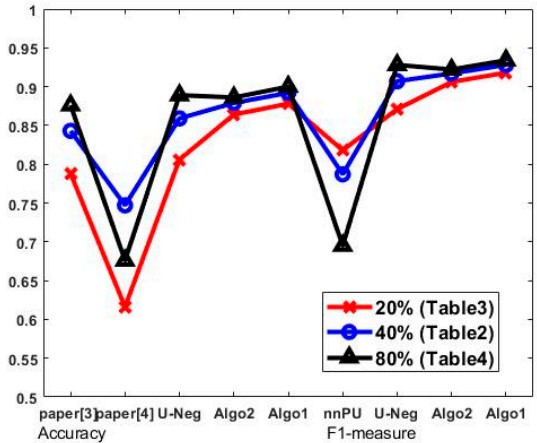

**Figure 2.** The comparison of average accuracy and F1-measure of the MPU and PU learning methods varying the size of unlabeled negative data samples.

### 4.4. Ablation Study

Next, we tested the parameter sensitivity of each method using BBC and Reuters data in an experimental setup in which 40% of data samples randomly selected from each class constituted unlabeled negative data. The left graph of the first row of Figure 3 compares the accuracy by the method of [3] when changing the λ value in the loss function of Equation (1), and the right graph shows the accuracy by the proposed Algorithm 1 for k values in KNN. The graph in the second row shows the accuracy by the method of [4] and the proposed Algorithm 1 for various learning rates and weight decay values, respectively. The last graph compares F1-measure by nnPU [19] for three base classifiers. The method in [4] and nnPU show high sensitivity to the weight decay value and the base classifier, respectively. On the other hand, in other cases the performance was quite stable.

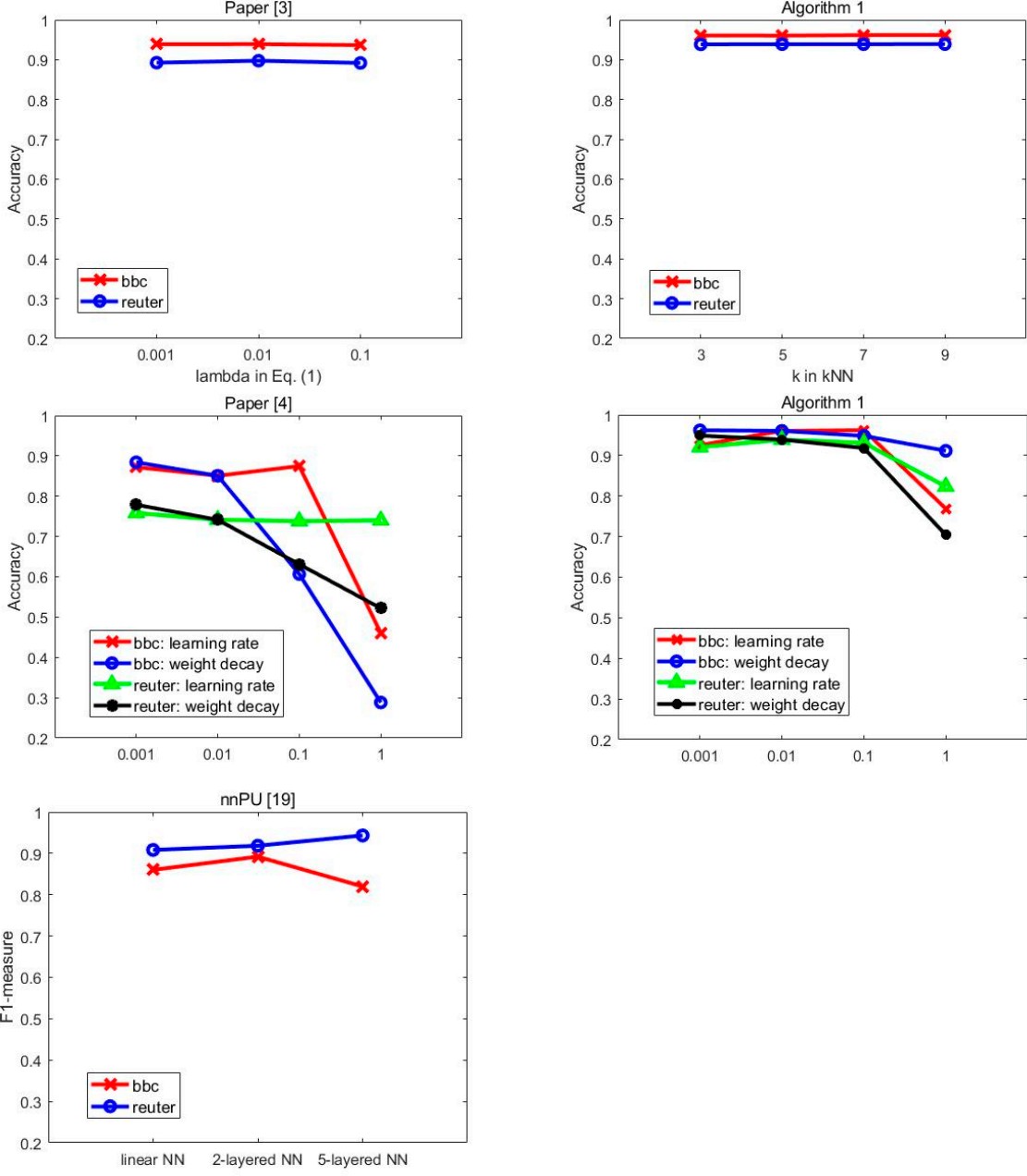

**Figure 3.** The performance comparison when changing parameter values.

## 5. Conclusions

Nowadays data of big size are easily collected as IT technology advances. However, data labeling needs a lot of effort and it is not guaranteed that unlabeled data would belong to one of the known classes. It can justify the necessity of classification algorithms utilizing unlabeled data which might belong to unknown classes. In this study, we proposed a method for multi-class positive and unlabeled learning on high dimensional data using outlier detection in a low dimensional embedding space. A multi-class classifier for $m - 1$ multiple positive classes and one negative class is constructed by a linear discriminant function that maps data to $m$ dimensional space. Selection of reliable negative samples based on outlier detection in the embedding space and the update of the linear discriminant function using positive data and the selected negative data are repeated alternately. Experimental results showed that the proposed method obtained a higher accuracy than the compared MPU learning methods. In particular, the performance of the proposed method was stable regardless of the amount of unlabeled negative data.

**Funding:** This work was supported by research fund of Chungnam National University.

**Conflicts of Interest:** The author declares no conflict of interest. The funders had no role in the design of the study; in the collection, analyses, or interpretation of data; in the writing of the manuscript; or in the decision to publish the results.

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
