# Peer review of "Multi-Class Positive and Unlabeled Learning for High Dimensional Data Based on Outlier Detection in a Low Dimensional Embedding Space"

_electronics, doi:10.3390/electronics11172789_

Round 1

Reviewer 1 Report (Previous Reviewer 3)

Now, everything looks good and is more informative. 

I would accept it in its present form. 

Author Response

Reviewer 2 Report (Previous Reviewer 2)

Content

----------

The goal of this paper is to propose a two-step approach for MPU learning on high dimensional data. In the first step, negative samples are selected from unlabeled data using an ensemble of k nearest neighbors-based outlier detection models in a low dimensional space which is embedded by a linear discriminant function.In the second step, the linear discriminant function is optimized on the labeled positive data and negative samples selected in the first step.

The author present the result alternates between updating the parameters of the linear discriminant function and selecting reliable negative samples by detecting outliers in a low-dimensional space.

The experimental results using high dimensional text data demonstrate the high performance of the proposed MPU learning method.

Major comments

--------------

1. 

Table 2. Performance comparison by accuracy and F1-measure.   

Table 3. Performance comparison by accuracy and F1-mesure when 20% data samples randomly selected from each class constitute unlabeled negative data.

Table 4. Performance comparison by accuracy and F1-measure when 80% data samples randomly selected from each class constitute unlabeled negative data.

There are many nnPU[19] in the result table. What is the relationship between mPU and nnPU ?

If this article focused on nnPU, the title should be changed to nnPU learning.

Evaluation

--------------

Given the above, I'm in a position to major revision.

Author Response

Reviewer 3 Report (Previous Reviewer 1)

All my questions were well addressed. It can be accepted now. 

Round 2

Reviewer 2 Report (Previous Reviewer 2)

The author addressed all my previous concerns.

Author Response

This manuscript is a resubmission of an earlier submission. The following is a list of the peer review reports and author responses from that submission.

Round 1

Reviewer 1 Report

The author proposed a two-step method to handle the MPU leaning. This method seems inspired by ref 4. Although the results prove the improved accuracy, the presentation needs to be improved. More additional results and ablation experiments should be designed and included to demonstrate the performance and effectiveness further. Some detailed comments are listed below.

1. In line 28, what does “…or new classes may emerge over time.” means?

2. What is the result in Ref.4? Please complete your comparison in lines 42-44.

3. Please modify the form of equations.

4. In lines 95-96, “may cause high complexity” is confusing.

5. In line 103, what does “(with bias)” means?

6. In Algorithm 1, please complete the explanation of Net(PU).

7. In lines 234-237, the authors used the default values for all methods. Yet, does the parameter optimization significantly affect the final performance of each approach? If not, why?

8. What the high dimension means? Table 1 shows that the dimension refers to the number of terms.

9. In Tables 2-4, what losses do you use for the results of Paper 3 and 4?

10. The quality of the figure is too low.

11. What is the accuracy of MPU, and how to calculate it? Are there some other indicators to evaluate the performance?

12. More ablation experiments are needed to prove the effectiveness of the proposed method. And the results presented are not convincing enough.

Reviewer 2 Report

Content

----------

The goal of this paper is to propose a two-step approach for MPU learning on high dimensional data. In the first step, negative samples are selected from unlabeled data using an ensemble of k nearest neighbors-based outlier detection models in a low dimensional space which is embedded by a linear discriminant function. The author present an approach for binary prediction which determines whether a data sample is a negative data sample. In the second step, the linear discriminant function is optimized on the labeled positive data and negative samples selected in the first step. It alternates between updating the parameters of the linear discriminant function and selecting reliable negative samples by detecting outliers in a low-dimensional space.

The experimental proposal is to use high dimensional text data demonstrate the high performance of the proposed MPU learning method.

Moreover, the author also shows the performance of the proposed method was stable regardless of the amount of unlabeled negative data.

Major comments

--------------

1. Figure 1. The comparison of average accuracies of the MPU learning methods varying the size of unlabeled negative data samples.

The qualify of the figure is low.

2. The author does not compare with other existing outlier detection methods.

Evaluation

--------------

Given the above, I'm in a position to major revision.

Reviewer 3 Report

It is quite an interesting paper, I do have two concerns:

1. Could you use a good quality image?

2. Is it possible to use any flow chart representation?